# In-Context Learning with Hypothesis-Class Guidance

## Abstract

Recent research has investigated the underlying mechanisms of in-context learning (ICL) both theoretically and empirically, often using data generated from simple function classes. However, the existing work often focuses on the sequence consisting solely of labeled examples, while in practice, labeled examples are typically accompanied by an *instruction*, providing some side information about the task. In this work, we propose *ICL with hypothesis-class guidance (ICL-HCG)*, a novel synthetic data model for ICL where the input context consists of the literal description of a (finite) hypothesis class $\mathcal{H}$ and $(x, y)$ pairs from a hypothesis chosen from $\mathcal{H}$. Under our framework ICL-HCG, we conduct extensive experiments to explore: (i) a variety of generalization abilities to new hypothesis classes; (ii) different model architectures; (iii) sample complexity; (iv) in-context data imbalance; (v) the role of instruction; and (vi) the effect of pretraining hypothesis diversity. As a result, we show that (a) Transformers can successfully learn ICL-HCG and generalize to unseen hypotheses and unseen hypothesis classes, and (b) compared with ICL without instruction, ICL-HCG achieves significantly higher accuracy, demonstrating the role of instructions.

## 1 Introduction

**LLMs and ICL**   Large Language Models (LLMs) (Zhao et al., 2023) have garnered widespread attention for their ability to solve complex tasks using simple text prompts. Among their many capabilities, in-context learning (ICL) (Brown et al., 2020) is particularly striking. ICL enables LLMs to adapt to new tasks by conditioning on provided examples, effectively allowing them to learn from context without explicit parameter updates. Understanding how such behavior emerges in LLMs remains an intriguing and challenging problem.

**Existing Efforts for Understanding ICL**   To elucidate the mechanisms behind ICL, researchers have constructed a variety of synthetic datasets (Garg et al., 2022; Li et al., 2023; Bai et al., 2023). These datasets typically involve sequences consisting of input-output pairs $\{(\boldsymbol{x}^{(i)}, y^{(i)})\}$, where each output $y^{(i)}$ is generated by a simple underlying function $f(\boldsymbol{x}^{(i)})$. For example, Garg et al. (2022) focus on noiseless linear regression, where each input is sampled from an isotropic Gaussian by $\boldsymbol{x}^{(i)} \sim \mathcal{N}(\boldsymbol{0}, \boldsymbol{I}_d)$, and the corresponding output is given by $y^{(i)} = \langle \boldsymbol{x}^{(i)}, \boldsymbol{w} \rangle$ with $\boldsymbol{w} \sim \mathcal{N}(\boldsymbol{0}, \boldsymbol{I}_d)$ for each sequence. During training and inference, the model receives a sequence consisting of $k$ demonstration pairs $(\boldsymbol{x}^{(1)}, y^{(1)}, \ldots, \boldsymbol{x}^{(k-1)}, y^{(k-1)})$ followed by a query input $\boldsymbol{x}_{\text{query}}$. This setup allows the model to infer the correct response for $\boldsymbol{x}_{\text{query}}$ conditioned on in-context examples. Various extensions have been proposed, including using Gaussian mixtures rather than a single Gaussian for task priors (Lin & Lee, 2024), employing non-linear functions (Bai et al., 2023), and introducing multiple intermediate "chain-of-thought" (Wei et al., 2022) steps within each $(\boldsymbol{x}, \boldsymbol{y})$ pair (Li et al., 2024a).

**Motivation** While a variety of data models have been studied to advance our understanding of ICL, a gap remains between these datasets and real-world ICL scenarios. In practice, users often provide LLMs with *an instruction* in addition to labeled demonstrations, containing the descriptions of the task in mind. See Fig. 1 for the visualization. In the top-left panel, the user provides a one-shot English-Korean translation pair without specifying the instruction, leading to an incorrect translation. In contrast, the top-right panel includes the instruction—*"perform the translation task following the demonstration"*—guiding the model to produce a correct translation, emphasizing the importance of the task descriptions. In fact, instructions are known to enhance the accuracy of ICL in general (Brown et al., 2020). However, most existing synthetic data

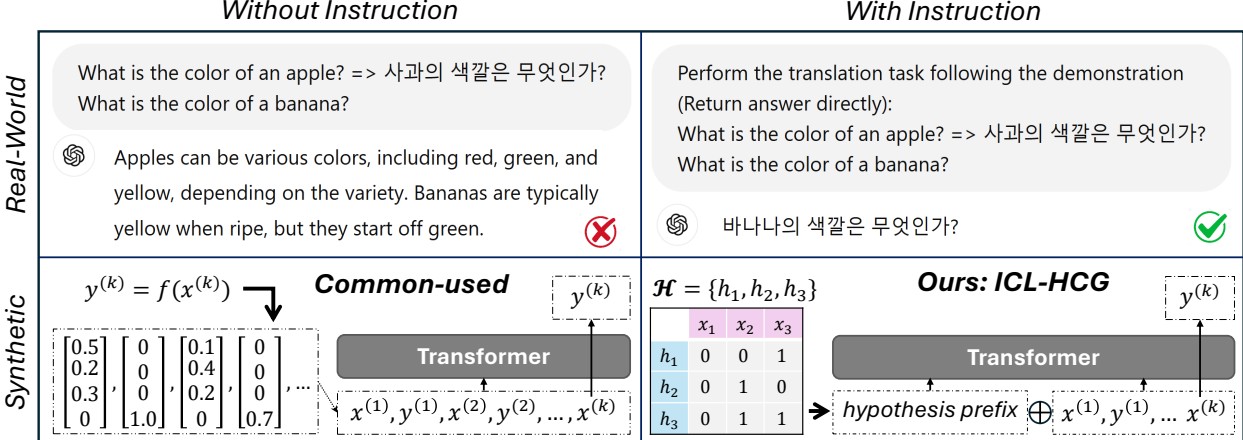

Figure 1: **Common ICL framework *vs.* ours.** Conventional frameworks with synthetic datasets often construct sequences by concatenating multiple $(\boldsymbol{x}, \boldsymbol{y})$ pairs, overlooking the importance of instructions. In contrast, our approach explicitly incorporates instructions through a *hypothesis prefix*. Specifically, we transform the hypothesis class $\mathcal{H}$ into a sequence that is prepended to the sequence of $(\boldsymbol{x}, \boldsymbol{y})$ pairs and then fed into a Transformer. We refer to this method as *in-context learning with hypothesis-class guidance (ICL-HCG)*. (Real-world examples are demonstrated using the GPT-4 Legacy model.)

frameworks overlook this crucial aspect, neglecting the role of instructions in guiding the learning process. Motivated by this limitation, we ask:

*Can we design a synthetic data framework for ICL that better captures the practical use scenarios of ICL by incorporating both instructions and labeled samples?*

Notably, two recent works (Xuanyuan et al., 2024; Huang & Ge, 2024) adopt prefix as instruction to implicitly provide information on the task. In contrast, our approach explicitly provides a hypothesis class as a prefix to the Transformer, guiding the model's understanding of the intended task.

**Our Synthetic Data Model** We propose a novel synthetic data model, *in-context learning with hypothesis-class guidance (ICL-HCG)*, illustrated in the bottom-right panel of Fig. 1, which integrates a hypothesis class into the ICL procedure. Specifically, besides the usual sequences of $(\boldsymbol{x}, \boldsymbol{y})$ pairs, a hypothesis class is embedded as a hypothesis prefix and fed into the Transformer (more details in Fig. 3 of Sec. 3.5). Leveraging this framework, we explore several aspects of Transformer behavior on the ICL-HCG task: (i) We evaluate the generalization ability of trained models to new hypothesis classes, new hypotheses, and various sizes of hypothesis classes; (ii) We compare different model architectures (Transformer, Mamba, LSTM, and GRU), highlighting their distinct properties on these generalizations; (iii) We examine the sample complexity required for achieving ID and OOD hypothesis class generalization and discover that merely a few dozen training hypothesis classes are sufficient for near-perfect generalization; (iv) We examine the effect of imbalanced in-context samples, demonstrating that imbalance can slow down the training process; (v) We assess the benefit of incorporating a hypothesis prefix, which notably enhances the accuracy of ICL; (vi) We show pretraining hypothesis diversity can significantly improve the accuracy of ICL when with instruction.

We summarize our contributions as follows:

- We propose a novel synthetic data model, namely in-context learning with hypothesis-class guidance (ICL-HCG) that integrates a hypothesis class into the ICL procedure. This design provides a controlled testbed for diverse experiments to study behaviors of ICL with instruction.

- We perform extensive empirical evaluations on our framework. Most interestingly, we demonstrate that (a) Transformers can successfully learn ICL-HCG and such a learned ability can generalize to unseen

hypotheses and unseen hypothesis classes, and (b) compared with ICL without instruction, ICL-HCG achieves significantly higher accuracy on ICL, demonstrating the role of instructions.

## 2 Related Works

Garg et al. (2022) first construct synthetic data with pretraining and testing sequences generated from noiseless linear regression tasks. Specifically, Garg et al. (2022) sample all input vectors $\boldsymbol{x}$ from an isotropic Gaussian distribution $\mathcal{N}(\boldsymbol{0}, \mathbf{I})$. Within each sequence, the outputs are given by $y^{(i)} = \langle \boldsymbol{w}, \boldsymbol{x}^{(i)} \rangle$, where $\boldsymbol{w}$ is drawn from $\mathcal{N}(\boldsymbol{0}, \mathbf{I}_d)$. For ICL inference, the prompt takes the form $(\boldsymbol{x}^{(1)}, y^{(1)}, \boldsymbol{x}^{(2)}, y^{(2)}, \ldots, \boldsymbol{x}^{(k-1)}, y^{(k-1)}, \boldsymbol{x}_{\text{query}})$ where the $k-1$ pairs $\{(\boldsymbol{x}^{(i)}, y^{(i)})\}_{i=1}^{k-1}$ serve as demonstrations illustrating the linear relationship governed by $\boldsymbol{w}$. The model then predicts the output for $\boldsymbol{x}_{\text{query}}$.

**Noiseless Linear Regression**    Based on the well-defined problem setup by Garg et al. (2022) using noiseless linear regression, researchers systematically study the mechanisms of ICL and properties of Transformer. For instance, there is a particular interesting line of research on connecting ICL to gradient descent, firstly hinted by Garg et al. (2022). Akyürek et al. (2023) and von Oswald et al. (2023) then show that one attention layer can be exactly constructed to perform gradient descent, and empirically find similarities between ICL and gradient descent. Further, Ahn et al. (2023) theoretically show that under certain conditions, Transformers trained on noiseless linear regression tasks minimizing the pretraining loss will implement gradient descent. Nevertheless, Fu et al. (2024) show that Transformers learn to approximate second-order optimization methods for ICL, sharing a similar convergence rate as Iterative Newton's Method. Besides gradient descent, there are lots of other interesting topics on ICL and Transformers based on this linear regression setting, such as looped Transformer (Yang et al., 2024; Gatmiry et al., 2024), training dynamic (Zhang et al., 2024; Huang et al., 2024; Kim & Suzuki, 2024), generalization (Panwar et al., 2024), etc.

**Noisy Linear Regression**    Such a simple noiseless linear regression task is further extended to variants. By extending the linear regression to noisy linear regression—$y = \langle \boldsymbol{x}, \boldsymbol{w} \rangle + \epsilon$, Li et al. (2023) analyze the generalization and stability of ICL. Wu et al. (2024) and Raventós et al. (2024) analyze the effect of task diversity on the attention model's ICL risk. Via extending the regression tasks sampling from Gaussian to Gaussian mixture, Lin & Lee (2024) show ICL exhibits two different modes including task retrieval and learning. With the tasks of $\boldsymbol{y} = \boldsymbol{W}\boldsymbol{x} + \boldsymbol{\epsilon}$ where $\boldsymbol{W}$ is a matrix rather than a vector, Chen et al. (2024) examine the training dynamic of multi-head attention for ICL.

**More than Linear Regression**    Beyond linear regression, researchers are also interested in non-linear regression and classification. The research directions are scattered, and we list them as follows. Bai et al. (2023) show that Transformers can perform in-context algorithm selection, *i.e.*, adaptively selecting different ICL algorithms such as gradient descent, least square, or ridge regression. Bhattamishra et al. (2024) show Transformers can learn a variety of Boolean function classes. Cheng et al. (2024) provide evidence that Transformers can learn to implement gradient descent to enable them to learn non-linear functions. Guo et al. (2024) show that trained Transformer achieves near-optimal ICL performance under $y = \langle \boldsymbol{w}, f(\boldsymbol{x}) \rangle$, where $f$ is a shallow neural network (MLP). Examining linear and non-linear regression tasks, Fan et al. (2024) and Tripuraneni et al. (2023) show Transformer can perform ICL on composited or mixed tasks of pretrained linear or non-linear tasks, and Yadlowsky et al. (2024) examine whether trained Transformers can generalize to new tasks beyond pretraining. Park et al. (2024) examine whether Mamba can in-context learn a variety of synthetic tasks. Via examining regression and classification tasks, Kim et al. (2024) show task diversity helps shorten the ICL plateau pretraining. Ramesh et al. (2024) assume there are multiple functions composited to connect $\boldsymbol{x}$ and $\boldsymbol{y}$ pair, *e.g.*, $\boldsymbol{y} = f_1 \circ f_2 \circ f_3(\boldsymbol{x})$ to study the compositional capabilities of Trasnformer. Li et al. (2024b) study how non-linear Transformer learns binary classification.

**Synthetic Dataset with Instruction**    To the best of our knowledge, there are two articles on synthetic datasets with instructions. Huang & Ge (2024) append an additional vector $\boldsymbol{\mu}$ to the sequences with interleaved input-output format, which leads to the sequence $(\boldsymbol{\mu}, \boldsymbol{x}^{(1)}, \boldsymbol{w}^\top \boldsymbol{x}^{(1)}, \boldsymbol{x}^{(2)}, \boldsymbol{w}^\top \boldsymbol{x}^{(2)}, \ldots)$ in which $\boldsymbol{x}^{(i)} \sim \mathcal{N}(\boldsymbol{\mu}, \boldsymbol{I})$, and show that the trained Transformer can achieve significantly lower loss on ICL when the task descriptor $\boldsymbol{\mu}$ is provided. The work of Xuanyuan et al. (2024) is most closely related to us. It develops a

new synthetic dataset based on task $((a \cdot x) \circ (b \cdot y)) \mod p = r$, where $(x, y)$ is the input, $r$ is the output, $\circ$ is an operation $(+, -, /)$, and each task is defined by the parameters $(a, b, \circ)$ ($p$ is a constant). The instruction is constructed as $(a_l, a_u, b_l, b_u, \circ)$, where $a_l$ and $a_u$ are the lower and upper bounds of $a$ (similar for $b$), and $\circ$ is the operation. Therefore, the instruction constrains the possible tasks, *i.e.*, provide information on the underlying true task of in-context samples. With such a setting, Xuanyuan et al. (2024) study how the information provided by instruction affects the ICL accuracy.

## 3 Meta-Learning for ICL-HCG

Training a learner to perform ICL aligns with the concept of meta-learning, as it enables adaptation to new tasks using in-context examples. While prior studies (Garg et al., 2022; Fan et al., 2024; Raventós et al., 2024) train Transformers for ICL on sequences of the form $(x_1, y_1, x_2, y_2, \ldots, x_k, y_k)$ without explicit instructions, our work investigates whether a Transformer trained for ICL with instructions, namely ICL-HCG, can generalize to new ICL-HCG tasks.

### 3.1 Two Types of Tasks in ICL-HCG

We consider two types of tasks in ICL-HCG, both constructed from a finite hypothesis class $\mathcal{H} = \{h^{(1)}, h^{(2)}, \ldots, h^{|\mathcal{H}|}\}$ over a finite input space $\mathcal{X} = \{x_1, x_2, \ldots, x_{|\mathcal{X}|}\}$ and a binary output space $\mathcal{Y} = \{0, 1\}$.

**Label prediction** Consider a hypothesis class $\mathcal{H}$ and a sequence of training data and a test point: $S_{k-1} \oplus x^{(k)} = (x^{(1)}, y^{(1)}, \ldots, x^{(k-1)}, y^{(k-1)}, x^{(k)})$, where for all $i$, $y^{(i)} = h(x^{(i)})$ for a specific $h \in \mathcal{H}$, and $x^{(k)}$ is a test query input. The objective is to predict: $y^{(k)} = h(x^{(k)})$. We refer to this as label prediction, with input-output pairs: $i_{\mathrm{I},k} = (\mathcal{H}, S_{k-1} \oplus x^{(k)})$ and $o_{\mathrm{I},k} = y^{(k)}$.

**Hypothesis identification** Given a hypothesis class $\mathcal{H}$ and a sequence (*ICL sequence*): $S_K = (x^{(1)}, y^{(1)}, \ldots, x^{(K)}, y^{(K)})$, where for all $i$, $y^{(i)} = h(x^{(i)})$ for a specific $h \in \mathcal{H}$, the goal is to identify the underlying hypothesis $h$. Denote this as hypothesis identification, with: $i_{\mathrm{II},K} = (\mathcal{H}, S_K)$ and $o_{\mathrm{II},K} = h$.

**Meta-learning** Label prediction uses $k-1$ samples to predict the label of a new query $x^{(k)}$, while hypothesis identification directly outputs $h$. Both label prediction and hypothesis identification can be viewed as attempts to identify $h$ from $\mathcal{H}$ via empirical risk minimization (ERM) using the dataset $\{(x^{(i)}, y^{(i)})\}$. Our meta-learning aims at learning to do ERM for different hypothesis classes when these hypothesis classes are given as input along with $(x, y)$ pairs.

### 3.2 Sample Generation

We consider the following two approaches for generating samples of ICL-HCG tasks.

**Assumption 1** (i.i.d. Generation)**.** *Given hypothesis classes $\{\mathcal{H}_i\}_{i=1}^N$, input space $\mathcal{X}$, and an integer $K$:*
*(a) Sample a hypothesis class $\mathcal{H}$ from $\{\mathcal{H}_i\}_{i=1}^{N^{train}}$;*
*(b) Sample a hypothesis $h$ uniformly at random from $\mathcal{H}$;*
*(c) Sample $K$ inputs $\{x^{(i)}\}_{i=1}^K$ i.i.d. from $\mathrm{Uniform}(\mathcal{X})$;*
*(d) Generate $y^{(i)} = h(x^{(i)})$ for each $i \in [K]$;*
*(e) $S_{k-1} \oplus x^{(k)} = [x^{(1)}, y^{(1)}, \ldots, x^{(k)}]$ for label prediction;*
*(f) $S_K = [x^{(1)}, y^{(1)}, \ldots, x^{(K)}, y^{(K)}]$ for hypothesis identification.*

**Assumption 2** (Opt-T Generation)**.** *Given hypothesis classes $\{\mathcal{H}_i\}_{i=1}^N$, input space $\mathcal{X}$, and an integer $K$:*
*(a) Sample a hypothesis class $\mathcal{H}$ from $\{\mathcal{H}_i\}_{i=1}^{N^{test}}$;*
*(b) Sample a hypothesis $h$ uniformly randomly from $\mathcal{H}$;*
*(c) Construct* optimal teaching set[1] *of $h$ with respect to $\mathcal{H}$;*
*(d) Randomly duplicate elements from this optimal teaching set until its size reaches $K$. Assign indices 1*

---

[1]The optimal teaching set (Zhu, 2015) is the smallest set of $(x, y)$ pairs that uniquely identifies $h$ among all candidates in $\mathcal{H}$.

*through $K$ arbitrarily to these $(x, y)$ pairs;*
*(e) $S_K = [x^{(1)}, y^{(1)}, \ldots, x^{(K)}, y^{(K)}]$ for hypothesis identification.*

### 3.3 Meta Training and Testing

**Training**  Given a set of training hypothesis classes $\{\mathcal{H}_i^{\text{train}}\}_{i=1}^{N^{\text{train}}}$, the meta-learner is trained in a multi-task setting to minimize the following loss:

$$\mathcal{L} = \mathcal{L}_1(f_\theta(i_{\text{II},K}), o_{\text{II},K}) + \sum_{k=1}^{K} \mathcal{L}_2(f_\theta(i_{\text{I},k}), o_{\text{I},k}), \tag{1}$$

where we generate $\mathcal{H}$, $h$, and $S_K$ following i.i.d. Generation, inherently defining $(i_{\text{II},K}, o_{\text{II},K})$ and $(i_{\text{I},k}, o_{\text{I},k})$. The loss is indeed implemented with additional terms, and we will further clarify the loss in Sec. 3.5, Eq. 2.

**Testing**  Given a set of testing hypothesis classes $\{\mathcal{H}_i^{\text{test}}\}_{i=1}^{N^{\text{test}}}$, we consider two types of testing.

- **Label prediction**: We generate $(i_{\text{I},k}, o_{\text{I},k})$ following i.i.d. Generation, and then measure whether the learner $f_\theta$ predicts $f(i_{\text{I},k})$ correctly for each $k \in [K]$;

- **Hypothesis identification**: We generate $(i_{\text{II},K}, o_{\text{II},K})$ using Opt-T Generation and evaluate whether the learner $f_\theta$ predicts $f(i_{\text{II}})$ correctly. This setting tests whether the learner acquires the ability to identify the underlying hypothesis with minimal information.

### 3.4 Four Types of Generalization

**Hypothesis universe $\mathcal{H}^{\text{uni}}$**  Given an input space $\mathcal{X} = \{x_1, x_2, \ldots, x_{|\mathcal{X}|}\}$ and a binary output space $\mathcal{Y} = \{0, 1\}$, We define the hypothesis universe $\mathcal{H}^{\text{uni}} = \mathcal{Y}^{\mathcal{X}}$ as the collection of all possible binary classification hypotheses. This universe contains $M = 2^{|\mathcal{X}|}$ distinct hypotheses, serving as a hypothesis pool to constructing training and testing hypothesis classes.

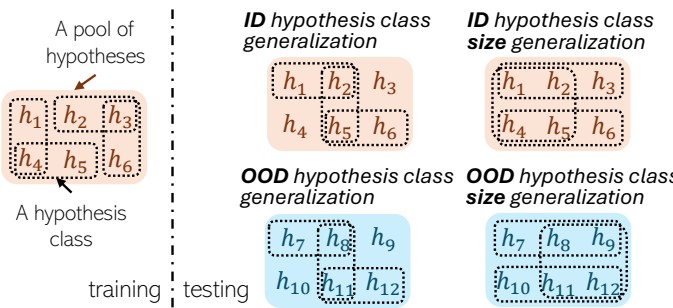

Figure 2: **Four types of generalization.** An illustration of the four types of generalization.

In meta-learning, the goal is to train a model that is able to rapidly adapt to new tasks. Testing on new tasks can be considered as measuring the OOD generalization. Under our ICL-HCG framework, we consider four types of OOD generalizations. First, we examine whether the learner generalizes to a new testing hypothesis class (the hypothesis class is unseen during training) that may or may not contain training hypotheses, referred to as in-distribution (ID) and out-of-distribution (OOD) hypothesis class generalization, respectively.

**Definition 3.1** (ID Hypothesis Class Generalization). Given $\mathcal{H}^{\text{uni}}$ of size $M$, we enumerate all $C(M, m) = \frac{M!}{m!(M-m)!}$ distinct hypothesis classes, each containing $m$ hypotheses. We then randomly *subsample* these classes into disjoint training and testing subsets, ensuring that no testing hypothesis class appears in the training set (although individual hypotheses may overlap). By training on randomly selected training hypothesis classes and evaluating on unseen testing hypothesis classes, we assess generalization to new hypothesis classes consisting of ID hypotheses.

**Definition 3.2** (OOD Hypothesis Class Generalization). Given $\mathcal{H}^{\text{uni}}$ of size $M$, we partition it into disjoint training and testing subsets of sizes $M^{\text{ID}}$ and $M^{\text{OOD}}$, respectively. We then generate training hypothesis classes from $M^{\text{ID}}$ and testing hypothesis classes from $M^{\text{OOD}}$, each containing $m$ hypotheses. We train the learner on the training hypothesis classes and evaluate on the testing hypothesis classes. Because no testing hypothesis appears during training, this setup probes how well the learner generalizes to entirely new hypotheses, *i.e.*, OOD hypotheses.

We then consider whether the learner can generalize to hypothesis classes of various sizes. Building on the concepts of ID and OOD hypothesis class generalization, we introduce size generalizations as follows.

**Definition 3.3** (ID Hypothesis Class Size Generalization). Based on ID hypothesis class generalization, while maintaining non-identical training and testing hypothesis classes, we allow the training hypothesis classes to have varying sizes. We investigate whether the learner generalizes well to testing classes of unseen sizes.

**Definition 3.4** (OOD Hypothesis Class Size Generalization). Based on OOD hypothesis class generalization, while maintaining non-identical training and testing hypotheses, we allow the training hypothesis classes to have varying sizes. We investigate whether the learner generalizes well to testing classes of unseen sizes.

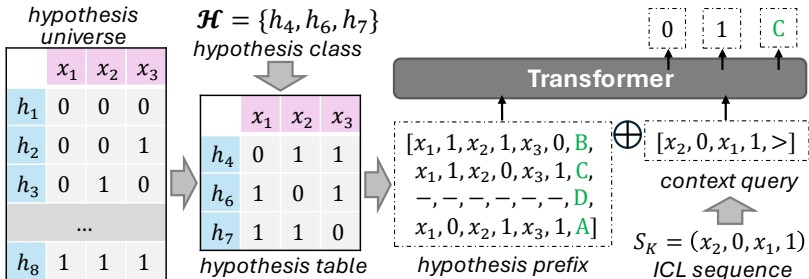

Figure 3: **Learning ICL-HCG via Transformer.** We begin by sampling a subset from the hypothesis universe as the hypothesis class $\mathcal{H}$. Next, we encode the hypothesis class $\mathcal{H}$ and concatenate it with the context query into a unified sequence of tokens. This sequence is fed into a Transformer model for training with next-token prediction, and testing for evaluating the accuracy on $y$'s and hypothesis identification. (This figure is a simplified illustration. Please refer to Appendix B and Fig. 13 for the full details.)

### 3.5 Learning ICL-HCG via Transformer

This section details how Transformer learns ICL-HCG. As shown in Fig. 3, the hypothesis class $\mathcal{H}$ is first converted to a hypothesis prefix with randomly assigned hypothesis indexes, then concatenated with context query representing sequence $S_K$ as a unified sequence $s$.

**Hypothesis prefix** [2] Given a hypothesis class $\mathcal{H} = \{h_4, h_6, h_7\}$, its hypothesis prefix with size $L = 4$ is constructed as shown in Fig. 3. Blank hypothesis is used to fill the hypothesis prefix when $|\mathcal{H}| < L$. A randomly assigned hypothesis index token $z$ is used to label each hypothesis. Leveraging Fig. 3 for $L = 4$, $z$'s are assigned from a pool {"A","B","C","D"} of size $L$ without replacement[3].

**Context query** Given an ICL sequence $S_K$, we append a query token ">" after it to trigger the prediction of the hypothesis index shown in Fig. 3. We name the combination of $S_K$ and ">" as context query.

The Transformer predicts the $y$ tokens in the context query based on previous tokens and the index $z$ of the underlying hypothesis based on all tokens in the sequence. The training loss in Eq. 1 is further extended to

---

[2] Please refer to Appendix B for the full version of hypothesis prefix.

[3] We use variable $z$ to represent the hypothesis index, and create a set of $L$ hypothesis index tokens as a pool from which each hypothesis is randomly assigned a unique index without replacement.

all the tokens in the sequence and implemented as below:

$$\mathcal{L} = -\sum_{t=1}^{T} \log P_\theta(s_i \mid s_{<i}).$$ (2)

We summarize the pipeline in the Appendix A Algorithm 1.

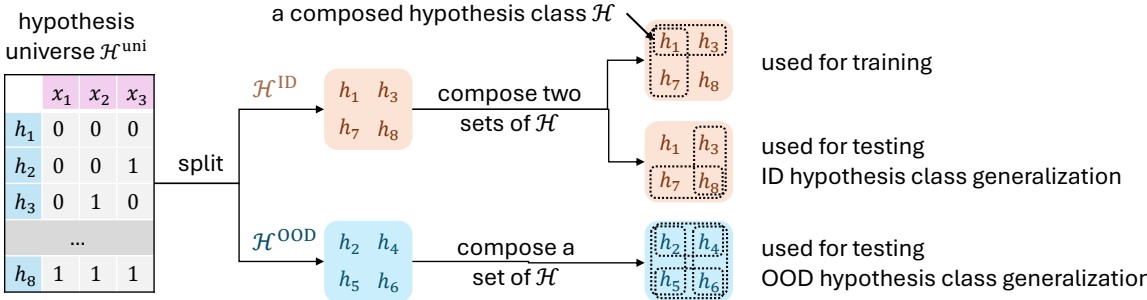

Figure 4: **The generation of training and testing hypothesis classes.** The hypothesis universe is partitioned into two pools: one for generating training and ID testing hypothesis classes, and another for generating OOD testing hypothesis classes.

## 4 Experiments

### 4.1 Setting of Experiments

**Hypothesis class generation**   Fig. 4 illustrates the hypothesis class generation process used in this paper. We partition the hypothesis universe into two pools: one for generating training and ID testing classes, and another for generating OOD testing hypothesis classes. This ensures that training and ID testing hypothesis classes do not overlap and that OOD hypothesis classes come from an entirely separate set of hypotheses. Consequently, both ID and OOD hypothesis class generalization can be assessed using the same trained model. For detailed realizations of setups for four kinds of generalization, see Appendix D.3.

**Pretraining**   During pretraining, we backpropagate gradients *based on next-token prediction for all tokens.* Each training sequence $s$ consists of a hypothesis prefix, a context query, and a hypothesis index token. As illustrated in Fig. 3, we feed the entire sequence $s$ excluding the final index token $z$ into the Transformer. We then compute cross-entropy loss for each subsequent token (excluding the very first). Refer to Appendix D for training details, including the learning rate schedule, and hyperparameter search.

**Components of pretraining loss**   We conducted experiments to determine the optimal components to include in the pretraining loss. Specifically, we evaluated four configurations: applying the loss (i) solely to the final hypothesis index token, (ii) exclusively to the content query, (iii) only to the label $y$ of the content query, and (iv) across all tokens. We empirically find that incorporating the loss across all tokens in the sequence leads to the best performance.

### 4.2 Four Types of Generalization

This section investigates whether a Transformer trained on ICL-HCG tasks can generalize to new tasks, *i.e.*, new hypothesis classes. We explore four types of generalization scenarios, defined in Definitions 3.1, 3.2, 3.3, and 3.4. Detailed hyperparameters of settings are provided in Appendix D.3.

> **Finding 1:**   *Transformer can successfully learn ICL-HCG tasks and such a learned ability can generalize to new hypothesis, hypothesis class, and hypothesis size, whereas the generalization on OOD hypotheses is harder than ID hypotheses.*

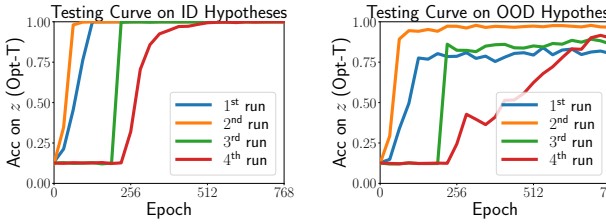

(a) testing curves of ID hypothe- (b) testing curves of OOD hy-
sis class generalization. pothesis class generalization.

Figure 5: **Multiple runs on ID and OOD hypothesis class generalizations.** (Different runs imply training and testing with different random seeds.) Transformer successfully learns ICL-HCG, and generalizes to new hypothesis classes and hypotheses. Generalization on ID hypotheses is easier than on OOD hypotheses. Refer to Appendix C.1, Fig. 14 for more curves of loss, training and testing accuracy.

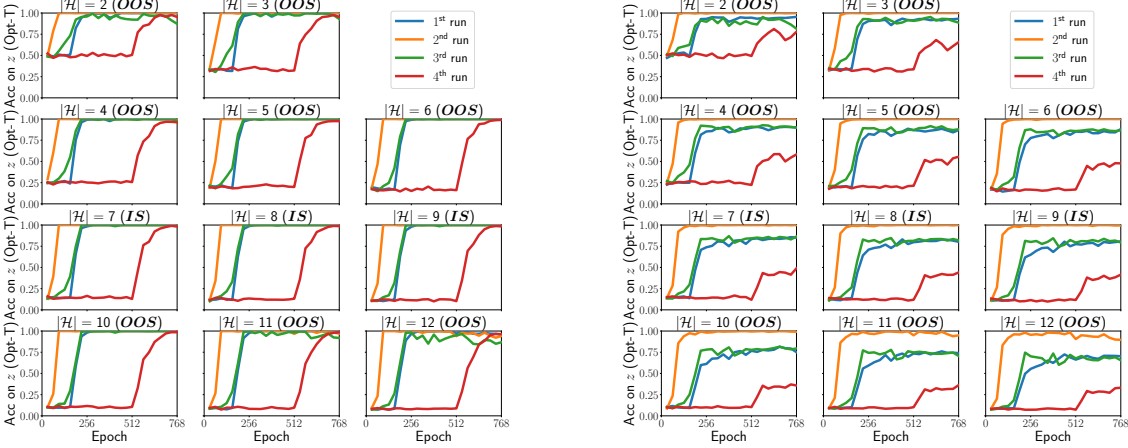

(a) Testing curves of ID hypothesis class size generalization. (b) Testing curves of OOD hypothesis class size generalization.

Figure 6: **Multiple runs on ID and OOD hypothesis class size generalizations.** (Different runs imply training and testing with different random seeds.) Transformers trained on hypothesis classes with sizes $|\mathcal{H}| \in \{7, 8, 9\}$ successfully generalizes to hypothesis classes with sizes $|\mathcal{H}| \in \{2, 3, \ldots, 13, 14\}$ under ID hypothesis class size generalization. In contrast, the same trained Transformer exhibits poorer performance on OOD hypothesis class size generalization. In the figure, ***IS* stands for "in-size,"** indicating the hypothesis class sizes included in the training, while ***OOS* stands for "out-of-size,"** indicating the sizes that are **not** included in the training. Refer to Appendix C.1, Fig. 15 for training accuracy curves.

We first demonstrate that the Transformer successfully learns ICL-HCG and that this capability generalizes effectively on ID and OOD hypothesis class generalizations. As illustrated in Figs. 5(a) and 5(b), the trained Transformers on 4 runs with different random seeds all achieve near-perfect accuracy (close to 1.00) on ID hypothesis class generalization, and around 0.8 to 0.9 accuracy on OOD hypothesis class generalization. Furthermore, we show that the learned ICL-HCG ability generalizes to hypothesis classes of various sizes. As depicted in Figs. 6(a) and 6(b), the trained Transformers achieve near 1.00 accuracy for $|\mathcal{H}| \in \{2, \ldots, 12\}$ on ID hypothesis class size generalization, while exhibiting moderately lower accuracy on OOD hypothesis class size generalization. Both Figs. 5 and 6 indicate that generalization on OOD hypotheses is more challenging compared to ID hypotheses.

### 4.3 Model Architecture Comparisons

We compare Transformer with other model architectures, including Mamba Gu & Dao (2023), LSTM Hochreiter (1997), and GRU Cho et al. (2014). We investigate whether each model can effectively fit the training dataset and, if so, generalize to the four types of unseen hypothesis classes.

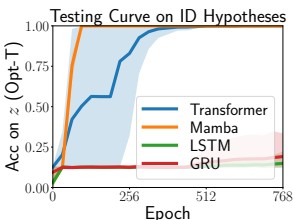 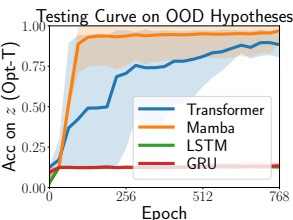

(a) testing curves of ID hypothesis class generalization.  (b) testing curves of OOD hypothesis class generalization.

Figure 7: **Various models on ID and OOD hypothesis class generalizations.** Transformer and Mamba succeed on both ID and OOD hypothesis class generalization, whereas LSTM and GRU fail. Mamba exhibits higher accuracy than Transformer on OOD generalization. See Appendix C.2 and Fig. 16 for training curves.

> **Finding 2:** *While both Mamba and Transformer excel on the four generalization tasks, LSTM and GRU fail to handle the ICL-HCG tasks. Mamba outperforms Transformer on OOD hypothesis class generalization, whereas Transformer outperforms Mamba on ID hypothesis class size generalization.*

We evaluate ID and OOD hypothesis class generalization across model architectures. Within the hyperparameter search space in Appendix D.2, Fig. 7 shows that Transformer and Mamba both generalize well on ID and OOD hypothesis class generalizations, with higher accuracy on ID hypotheses (1.00 accuracy) than OOD (around 0.8 to 0.9 accuracy). In contrast, LSTM and GRU fail to fit the task, achieving approximately 0.125 accuracy, equivalent to random guessing over eight hypotheses. Furthermore, Fig. 8 shows that Mamba outperforms Transformer on OOD hypothesis class size generalization, whereas Transformer excels on ID hypothesis class size generalization, suggesting a potential advantage of Transformer on length generalization, and Mamba on generalization of OOD hypotheses.

### 4.4 Effect of Training Hypothesis Class Count

We evaluate how the number of training hypothesis classes affects ID and OOD hypothesis class generalization.

> **Finding 3:** *Mamba is more sample efficient than Transformer on ICL-HCG tasks, and achieves near perfect generalization with only several pretraining hypothesis classes.*

In Fig. 9(a), we evaluate Mamba, Transformer, GRU, and LSTM. With only $2^2$ and $2^4$ training hypothesis classes, Mamba and Transformer achieve near-perfect (1.00 accuracy) ID hypothesis class generalization, while LSTM and GRU fail to fit the ICL-HCG tasks. In Fig. 9(b), Mamba nearly achieves perfect OOD hypothesis class generalization with as few as $2^2$ training classes, whereas Transformer's accuracy improves gradually with more training classes.

### 4.5 Effect of Imbalanced In-Context Samples

This section investigates how an imbalanced sample distribution in the context query affects the training procedure. Specifically, we consider the following distribution over $\mathcal{X}$:

$$\mathrm{norm}\left(\frac{1}{\sqrt{D}}, \ldots, \frac{1}{\sqrt{D}}, 1, \sqrt{D}, \ldots, \sqrt{D}\right),$$

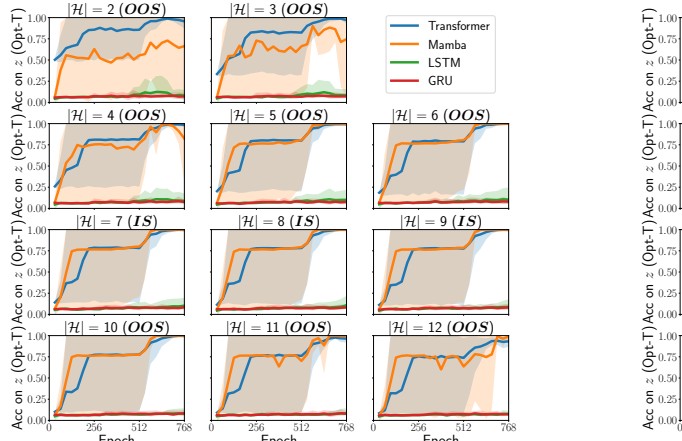 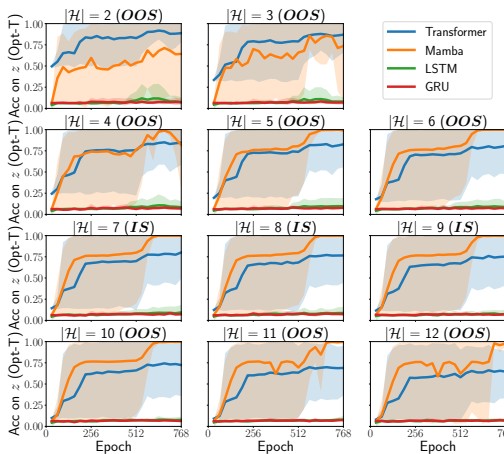

(a) testing curves of ID hypothesis class size generalization. (b) testing curves of OOD hypothesis class size generalization.

Figure 8: **Various models on ID and OOD hypothesis class size generalizations.** In both settings, Transformer and Mamba exhibit strong generalization, whereas LSTM and GRU fail to do so. For hypothesis class sizes $|\mathcal{H}| \in \{7, 8, 9\}$, Mamba achieves accuracy comparable to Transformer on ID hypothesis class generalization, and surpasses Transformer on OOD hypothesis class generalization. However, Transformers show similar or higher accuracy than Mamba on ID hypothesis class size generalization, suggesting a potential advantage in length generalization. Refer to Appendix C.2, Fig. 17 for training accuracy curves.

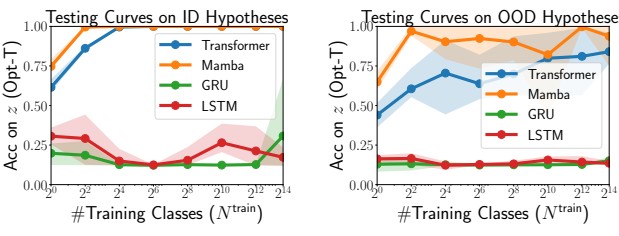

(a) testing curves of ID hypothe-(b) testing curves of OOD hy-
sis class generalization. pothesis class generalization.

Figure 9: **Effect of training hypothesis class count.** Transformer and Mamba trained on ICL-HCG tasks generalize to new hypothesis classes with only 4 to 16 training hypothesis classes. Refer to Appendix C.3, Fig. 18 for training accuracy and more details.

where the first half of the terms are $\frac{1}{\sqrt{D}}$, the middle term (if $|\mathcal{X}|$ is odd) is 1, the second half consists of $\sqrt{D}$, and $D^4$ represents the disparity of the distribution over $\mathcal{X}$, *i.e.*, $D = \frac{\max_{x \in \mathcal{X}} P(x)}{\min_{x \in \mathcal{X}} P(x)}$.

> **Finding 4:** *In-context sample imbalance lags the convergence of training.*

We analyzed the impact of imbalance on the training process in Fig. 10 by varying D values, showing that greater imbalance slows convergence. On average over four runs, training converges in about 384 epochs for $D = 1$ but takes around 700 epochs for D= 4.

---

[4]The notation D is distinguished from token "D" by color and dataset $\mathcal{D}$ by format.

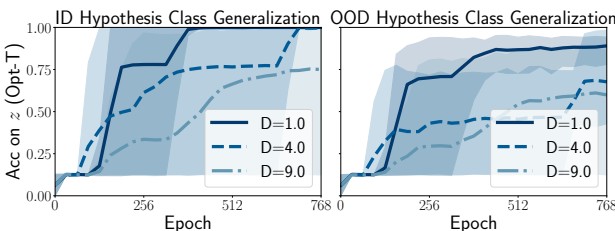

Figure 10: **The effect of sample imbalance.** Sample imbalance leads to lower convergence speed.

## 4.6 The Benefit of Hypothesis Prefix

In this section, we demonstrate how the hypothesis prefix influences the accuracy of ICL. We compare ICL accuracy on $y$ with hypothesis prefix and without hypothesis prefix, under ID hypothesis class generalization.

> **Finding 5:** *Incorporating hypothesis prefix as instruction significantly boost the accuracy of ICL.*

As shown in Fig. 11, the hypothesis prefix significantly enhances the training and testing accuracy on $y$ of ICL. Using position 3 as an example, predictions with three $(x, y)$ pairs as demonstrations achieve approximately 0.95 accuracy with instruction but only around 0.8 without, highlighting the effectiveness of instruction.

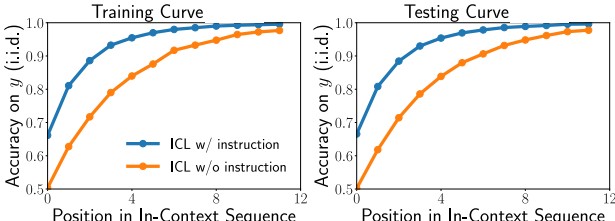

Figure 11: **The effect of instruction.** Under ID hypothesis class generalization, providing an instruction (hypothesis prefix) significantly boosts ICL performance, especially for $y$ token with small index (indicating only a few demonstration examples precede it).

## 4.7 The Effect of Pretraining Hypothesis Diversity

In this section, we investigate the impact of hypothesis diversity combined with instruction (hypothesis prefix) on ICL accuracy. We conduct experiments under OOD hypothesis class generalization with an input space size of $|\mathcal{X}| = 6$, leading to a hypothesis universe $\mathcal{H}^{\mathrm{uni}}$ of $2^{|\mathcal{X}|} = 64$ hypotheses. $\mathcal{H}^{\mathrm{uni}}$ is split into $\mathcal{H}^{\mathrm{ID}}$ with 48 hypotheses and $\mathcal{H}^{\mathrm{OOD}}$ with 16. For training, we sample $M^{\mathrm{train}} \in \{8, 16, 24, 32\}$ hypotheses from $\mathcal{H}^{\mathrm{ID}}$ to examine the effect of training hypothesis diversity, while testing uses all hypotheses in $\mathcal{H}^{\mathrm{ID}}$.

> **Finding 6:** *Increasing the diversity of pretraining hypotheses significantly boosts the performance of ICL when instructions are provided.*

As illustrated in Fig. 12, under OOD hypothesis class generalization, the Transformer trained with instructions achieves similar ICL accuracy to a standard ICL approach when pretraining hypothesis diversity is low, but significantly outperforms it when pretraining hypothesis diversity is high. Using position 10 as an example, with instruction, increasing $M^{\mathrm{train}}$ from 8 to 32 raises accuracy from 0.80 to approximately 0.99. Without instruction, the same increase in diversity improves accuracy only from 0.80 to 0.90. Notably, the testing ICL samples are derived from unseen hypotheses, indicating that incorporating instructions can enhance ICL performance for new hypotheses.

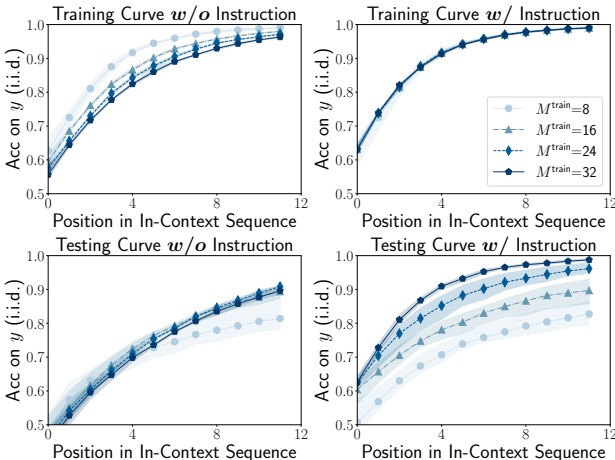

Figure 12: **The effect of pretraining hypothesis diversity.** Under hypothesis generalization, increasing the diversity of pretraining hypotheses significantly boosts the performance of ICL when instructions are provided. However, without instructions, this effect is limited.

## 5 Discussion

Building on the ICL-HCG framework, we conduct diverse experiments focusing on generalization. While Bayesian inference (Xie et al., 2021) offers insights into ICL, prior work (Raventós et al., 2024) has shown that Transformers can generalize beyond Bayesian inference with sufficient pretraining task diversity. However, the mechanisms underlying such OOD generalization remain unclear. Our work provides a framework for exploring OOD generalization beyond Bayesian inference, where no test samples appear in the training set due to disjoint hypothesis classes.

Furthermore, in Sec. 4.3, we demonstrate that Transformer and Mamba exhibit distinct strengths: Transformer excels on length generalization, while Mamba performs better on OOD hypothesis generalization. In Sec. 4.7, we show that instruction enhances the benefits of pretraining hypothesis diversity. These findings highlight two key factors influencing OOD generalization: (i) model architecture and (ii) data structure. Future work will further explore these phenomena, focusing on understanding the underlying mechanisms of OOD generalization in Transformer and Mamba.

## 6 Conclusion

In this paper, we introduced a novel instruction-based ICL framework that explicitly integrates a hypothesis class as the instruction, namely ICL-HCG. Through a series of diverse experiments, we demonstrated that Transformers trained on ICL-HCG tasks can generalize to new hypothesis classes and new hypotheses, even when trained on only a few hypothesis classes. Moreover, we show that the incorporation of such instructions significantly enhances the accuracy of ICL, thereby bridging the gap between synthetic ICL studies and real-world applications. We also examined the effect of hypothesis diversity within this framework and found that increased hypothesis diversity substantially improves ICL accuracy especially when instructions are provided. Our framework serves as a platform for diverse explorations, offering new insights and serving as a playground for research on ICL and LLMs.

We conclude our paper by acknowledging the limitations of our current framework: (i) Our study is confined to finite hypothesis binary classification problems, which can be extended to more complex scenarios; (ii) The hypothesis prefix is assumed to provide an explicit hypothesis class, differing from the more implicit instructions used in real-world LLM applications.

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

## A   Pseudo Algorithm for ICL-HCG

We summarize our meta framework for ICL-HCG in Algorithm 1.

---

**Algorithm 1** Meta-Learning Framework for ICL-HCG

---

1: **Inputs:** a set of inputs $\mathcal{X}$, a training set of hypothesis classes $\mathcal{S}^{\text{train}} = \{\mathcal{H}_i^{\text{train}}\}_{i=1}^{N^{\text{train}}}$, a testing set of hypothesis classes $\mathcal{S}^{\text{test}} = \{\mathcal{H}_i^{\text{test}}\}_{i=1}^{N^{\text{test}}}$, batch size $B$, hypothesis prefix size $L$, and context query size $K$
2: **for training epoch do**
3:     sample $\{\mathcal{H}_i\}_{i=1}^{B} \overset{\text{i.i.d.}}{\sim} \text{Uniform}(\mathcal{S}^{\text{train}})$
4:     **for** each hypothesis class $\mathcal{H} \in \{\mathcal{H}_i\}_{i=1}^{B}$ **do**
5:         generate $h, S_K$ following i.i.d. Generation
6:         **// Construct sequence based on $\mathcal{H}$, $h$, and $S_K$**
7:         construct hypothesis prefix, context query, and hypothesis index $z$ based on $\mathcal{H}, h, S_K$
8:         $s \leftarrow \text{concatenate}(\text{hypothesis prefix}, \text{context query}, z)$
9:         **// Cross-entropy loss for next token prediction**
10:         $\mathcal{L} \leftarrow -\sum_{t=2}^{|s|} \log P(s_t \mid s_{<t})$
11:     **end for**
12:     update model parameters using $\mathcal{L}$ of the batch
13: **end for**
14: **for testing epoch do**
15:     sample $\{\mathcal{H}_i\}_{i=1}^{B} \overset{\text{i.i.d.}}{\sim} \text{Uniform}(\mathcal{S}^{\text{test}})$
16:     **for** each hypothesis class $\mathcal{H} \in \{\mathcal{H}_i\}_{i=1}^{B}$ **do**
17:         generate $h, S_K$ via:
18:             **either** following i.i.d. Generation
19:             **or** following Opt-T Generation
20:         **construct sequence $s$ based on $\mathcal{H}$, $h$, and $S_K$**
21:         **evaluate the prediction accuracy on $y$, $z$, etc**
22:     **end for**
23: **end for**

---

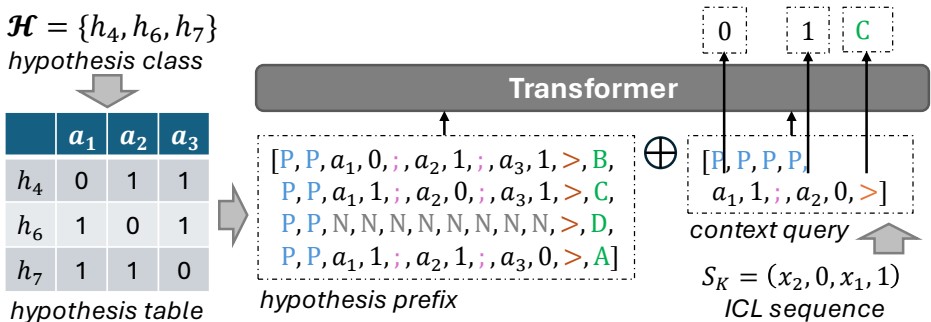

Figure 13: **The framework.** We convert hypothesis class $\mathcal{H}$ and ICL sequence $S_K$ into sequences of tokens, concatenate them and input to Transformer. Then we examine whether Transformer can predict correct $y$ and $z$ values.

## B  Implementation Detail of Hypothesis Prefix and Context Query

**Hypothesis prefix**   Given a hypothesis class $\mathcal{H}$ and its hypothesis table, the correspongding hypothesis prefix with hypothesis prefix's content length $L$ is constructed as shown in Fig. 13. The token "P" serves as the padding token to separate hypotheses, the token ";" serves as the separation token to separate $(x, y)$ pairs, the token "N" serves as the empty token to fill a blank hypothesis, and the token ">" is used to connect $(x, y)$ pairs of the hypothesis to a randomly assigned hypothesis index $z$[5]. In the illustrated example in Fig. 13, the randomly assigned indexes $z$'s are sampled from $M = 4$ hypothesis index tokens {"A","B","C","D"} without replacement[6].

**Context query**   Given an ICL sequence $S_K$ with $K$ pairs of $(x, y)$, the context query of size $K$ is constructed to represent the ICL sequence and trigger the prediction of the hypothesis index with padding token "P", separation token token ";", and query token ">" as shown in Fig. 13.

## C  Additional Details of Experiments

### C.1  Four Types of Generalization

We share more training and testing curves in Fig. 14 to provide additional results to Fig. 5, and in Fig. 15 to provide additional results to Fig. 6.

### C.2  Compare with Other Model Architectures

We share more training and testing curves in Figs. 16 and 17 to provide additional results to Figs. 7 and 8, respectively.

### C.3  Effect of Training Class Count

We share more training and testing curves in Fig. 18 to provide additional results to Fig. 9.

## D  Experiment Setup

Each experiment is **repeated four times**, with the mean calculated across runs. The shadow region's boundary is defined by **the minimum and maximum values** observed across the four runs.

---

[5]We use variable $z$ to represent the hypothesis index.

[6]A set of $L$ hypothesis index tokens are created serve as the pool from which the hypothesis indexes are randomly sampled without replacement.

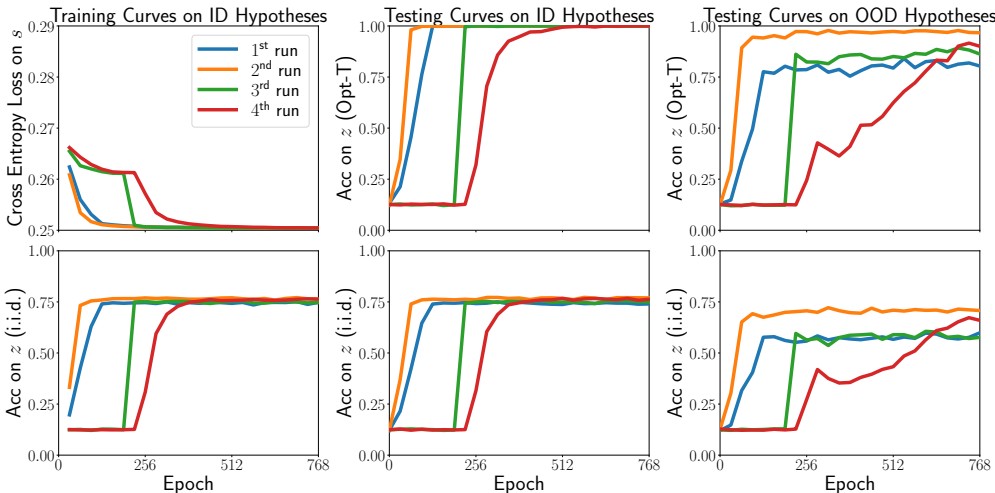

Figure 14: **Multiple runs for ID and OOD hypothesis class generalizations.**

## D.1 Learning Rate Scheduler

We set the train procedure with 768 total epochs, each epoch containing 1024 batches. The learning rate (lr) is first warmed up linearly from an LR/64 at epoch $e = 1$ to a peak value LR at epoch $e = 64$, following:

$$\text{lr}(e) = \text{LR} \cdot \frac{e}{64}, \quad 1 \le e \le 64.$$

After epoch 64, the learning rate undergoes a quadratic decay over the remaining 704 epochs, given by

$$\text{lr}(e) = \text{LR} \cdot \sqrt{\frac{64}{e}}, \quad 64 \le e \le 768.$$

## D.2 Hyperparameter Search

We list the hyperparameter searching spaces used for Transformer, LSTM, GRU, and Mamba. The best hyperparameter is searched using ID hypothesis class generalization with $\|\mathcal{X}\| = 5$, $\|\mathcal{H}\| = 8$, and then used for all other settings.

Table 1: **Hyperparamer searching spaces for different model architectures.** The optimal hyperparameters are bolded if multiple possibilities are provided.

| Model Architecture | #layers | #hidden dimensions | #learning rate | #weight decay | #batch size |
|---|---|---|---|---|---|
| Transformer | **2**,8 | 128 | 0.00010, **0.00020**, 0.00050, 0.00100 | 0.0005 | 16 |
| Mamba | **2**,8 | 128 | 0.00010, 0.00020, **0.00050**, 0.00100 | 0.0005 | 16 |
| GRU | **2**,8 | 128 | 0.00020, 0.00050, **0.00100**, 0.00200 | 0.0005 | 16 |
| LSTM | **2**,8 | 128 | 0.00020, 0.00050, **0.00100**, 0.00200 | 0.0005 | 16 |

## D.3 Setup for Generating Training and Testing Hypothesis Classes

We list the experimental setup for each experiments in the following Table 2. When conducting experiments to evaluate accuracy on $y$, we modified the experimental setup following Table 3.

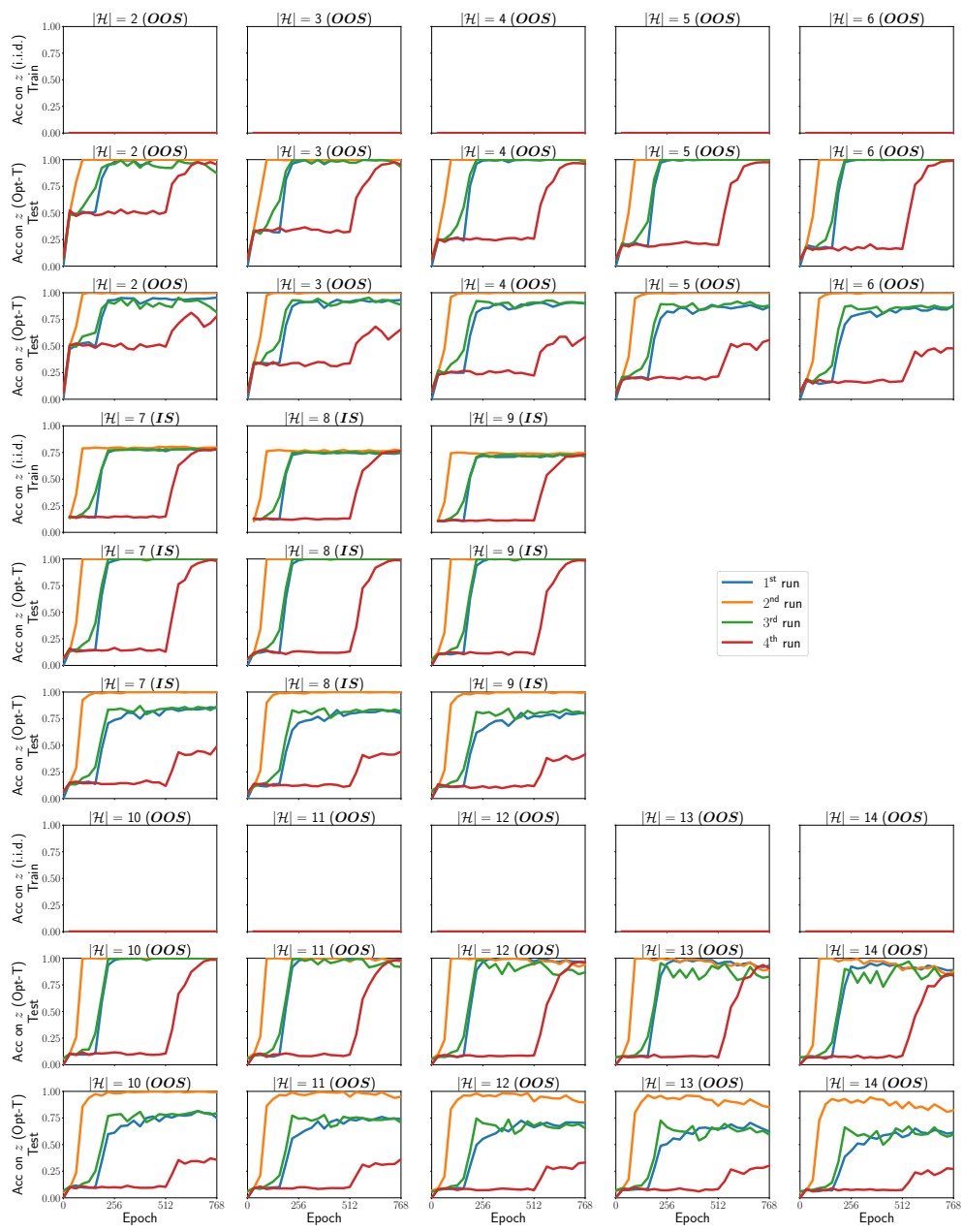

Figure 15: **Multiple runs for ID and OOD hypothesis class size generalizations.**

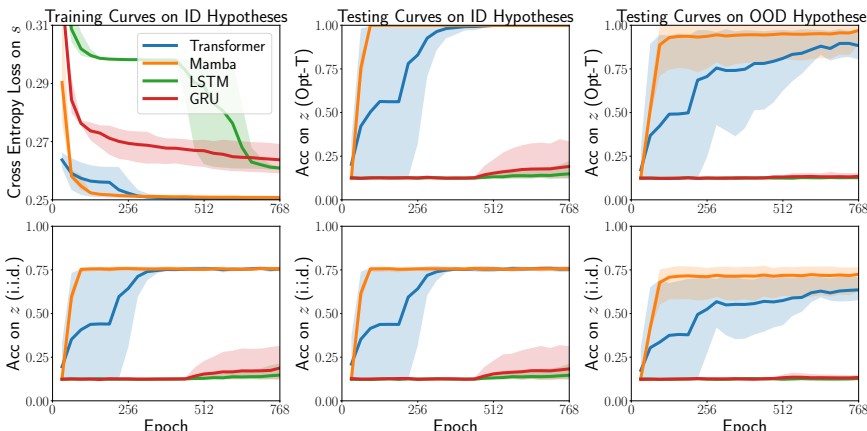

Figure 16: **Various models on ID and OOD hypothesis class generalizations.**

Table 2: **Experimental setups of different generalizations.** The expression $\min\{512, \#\text{possible}\}$ indicates that when the number of possible hypothesis classes is fewer than 512, we evaluate all possible hypothesis classes for testing; otherwise, we limit the selection to at most 512 hypothesis classes. For example, if $|\mathcal{H}^{\text{OOD}}| = 16$ and $|\mathcal{H}| = 2$, the total number of possible hypothesis classes is given by: $\binom{|\mathcal{H}^{\text{OOD}}|}{|\mathcal{H}|} = \binom{16}{2} = \frac{16 \times 15}{2} = 120$. Since $120 < 512$, we evaluate all 120 hypothesis classes for testing in this scenario.

| Generalization Setup | ID Hypothesis Class Generalization | OOD Hypothesis Class Generalization | ID Hypothesis Class Size Generalization | OOD Hypothesis Class Size Generalization |
|---|---|---|---|---|
| size of input space ($|\mathcal{X}|$) | 5 | 5 | 5 | 5 |
| size of label space ($|\mathcal{Y}|$) | 2 | 2 | 2 | 2 |
| size of context query ($K$) | 5 | 5 | 5 | 5 |
| size of training hypothesis class ($|\mathcal{H}^{\text{train}}|$) | 8 | 8 | 7,8,9 | 7,8,9 |
| size of testing hypothesis class ($|\mathcal{H}^{\text{test}}|$) | 8 | 8 | 2,...,14 | 2,...,14 |
| size of hypothesis prefix ($L$) | 8 | 8 | 16 | 16 |
| #all hypotheses ($|\mathcal{H}^{\text{uni}}|$) | 32 | 32 | 32 | 32 |
| #hypotheses in ID pool ($|\mathcal{H}^{\text{ID}}|$) | 16 | 16 | 16 | 16 |
| #hypotheses in OOD pool ($|\mathcal{H}^{\text{OOD}}|$) | 16 | 16 | 16 | 16 |
| #training hypothesis classes | 12358 | 12358 | 4096 | 4096 |
| #testing hypothesis classes | 512 | 512 | $\min\{512, \#\text{possible}\}$ | $\min\{512, \#\text{possible}\}$ |

Table 3: **Additional setups.** Numbers that differ from those in Table 2 are highlighted in bold for clarity.

| Section | Sec. 4.6 | Sec. 4.7 |
|---|---|---|
| size of input space ($|\mathcal{X}|$) | **4** | **6** |
| size of label space ($|\mathcal{Y}|$) | 2 | 2 |
| size of context query ($K$) | **12** | **12** |
| size of training hypothesis class ($|\mathcal{H}^{\text{train}}|$) | **4** | 8 |
| size of testing hypothesis class ($|\mathcal{H}^{\text{test}}|$) | **4** | 8 |
| size of hypothesis prefix ($L$) | **4** | 8 |
| #all hypotheses ($|\mathcal{H}^{\text{uni}}|$) | **16** | **64** |
| #hypotheses in ID pool ($|\mathcal{H}^{\text{ID}}|$) | **16** | **8,16,24,32,48** |
| #hypotheses in OOD pool ($|\mathcal{H}^{\text{OOD}}|$) | **0** | 16 |
| #training hypothesis classes | **1308** | $\min\{12358, \#\text{possible}\}$ |
| #testing hypothesis classes | 512 | 512 |

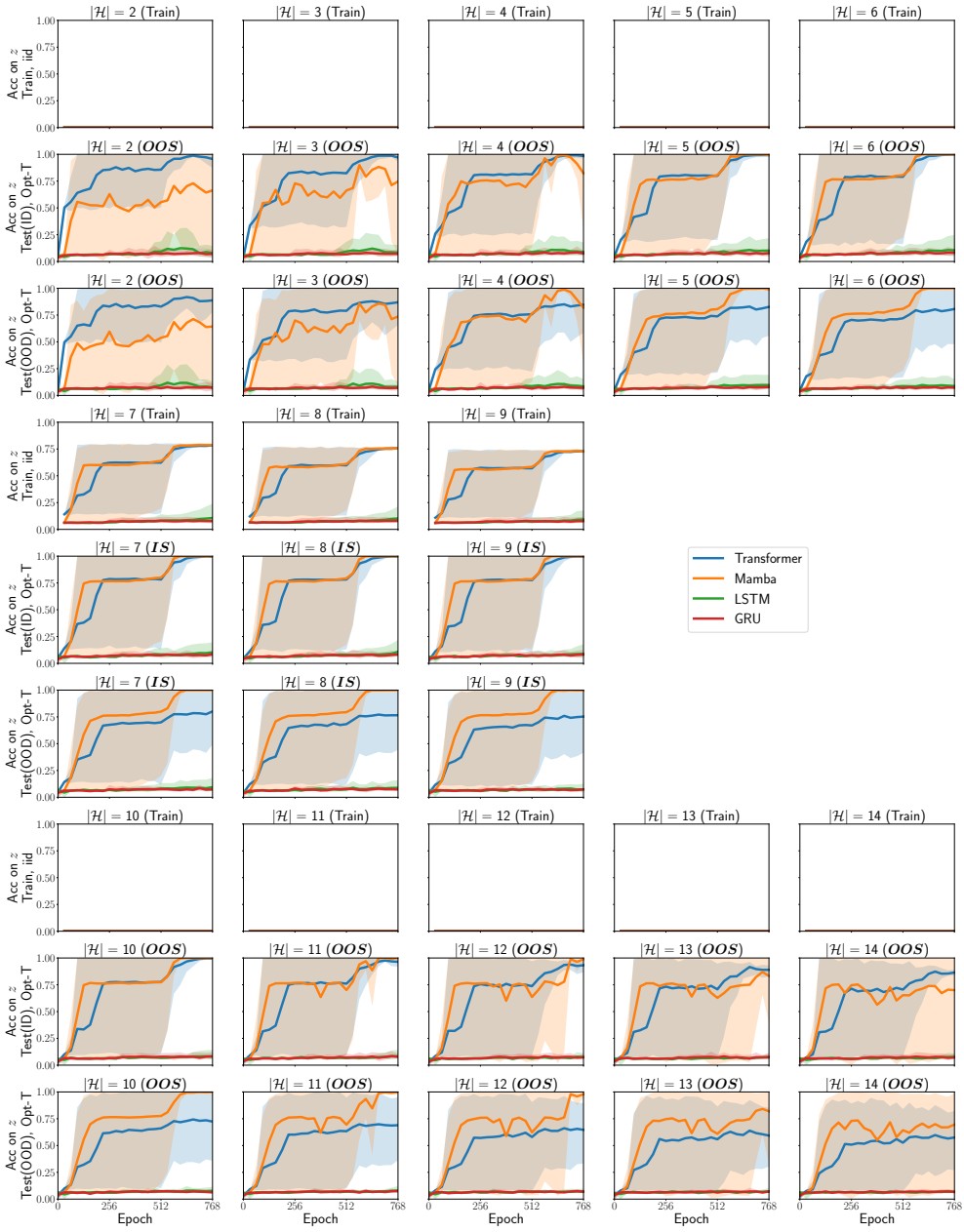

Figure 17: **Various models on ID and OOD hypothesis class generalizations.**

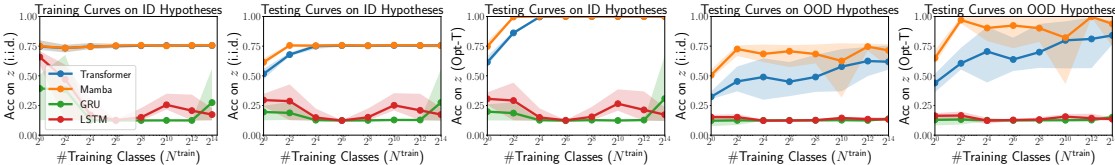

Figure 18: **Effect of training hypothesis class count on ID and OOD hypothesis class generalization.**

