# OpenReview forum: "In-Context Learning with Hypothesis-Class Guidance"
_TMLR — Rejected by TMLR_

### Review · Reviewer_5Gsi · 2025-07-21

**Summary Of Contributions:**

The paper propose a novel synthetic data model, namely in-context learning with hypothesis-class guidance (ICL-HCG) that integrates a hypothesis class into the ICL procedure. This design provides a controlled testbed for diverse experiments to study behaviors of ICL with instruction.  The paper finds that (a) Transformers can successfully learn ICL-HCG and such a learned ability can generalize to unseen hypotheses and unseen hypothesis classes, and (b) compared with ICL without instruction, ICL-HCG achieves significantly higher accuracy on ICL, demonstrating the role of instructions.

**Audience:**

Yes

**Claims And Evidence:**

Yes

**Requested Changes:**

To strengthen the empirical validity of the proposed ICL-HCG framework, the paper would benefit from additional ablations and comparisons. In particular, key missing ablations include training with only the hypothesis prefix (without in-context examples), testing the robustness of prefix to noise, and exploring the tradeoff between prefix length and the number of in-context examples under a fixed token budget. Moreover, comparisons with more baselines are needed, such as recent methods that implicitly inject task information via symbolic prefixes ([1][2]), as well as natural language instructions as prefix alternatives, to better situate the proposed approach within ICL and instruction learning.

**Strengths And Weaknesses:**

1. Novelty:
This work is the first to explicitly present candidate hypotheses as Transformer prefixes and leverage random indexing to enable internal contrastive retrieval.

2. Comprehensiveness:
It systematically defines and empirically validates the four scenarios arising from ID/OOD hypothesis‐class shifts and sample‐size generalization, providing a structured perspective on in‐context generalization.

Weakness:
1. Need more ablation:

- No in-context examples, only hypothesis prefix during training.
- The robustness of the hypothesis prefix to noise
- The tradeoff between prefix length and number of in-context examples under token constraints
2. Need more baseline:

- Methods that implicitly provide task information via prefix ([1][2])

[1] On Task Description of In-context Learning: A Study from Information Perspective
[2] Task Descriptors Help Transformers Learn Linear Models In-Context

- Natural language instruction as prefix

---

> ### Author Response · Authors · 2025-08-25
>
> > Need more ablation:
>
> A: We appreciate the reviewer for bringing these points to make our experiments more diverse and comprehensive. We will consider adding these experiments.
>
> >Need more baseline
>
> A: We have discussed the two papers in the related work. Our paper and these two papers work on different frameworks, which makes "comparing our framework with baselines on some benchmarks" not feasible. We will discuss more about the difference between our work and baselines.
>
> Requested Changes:
> > To strengthen the empirical validity of the proposed ICL-HCG framework, the paper would benefit from additional ablations and comparisons. In particular, key missing ablations include training with only the hypothesis prefix (without in-context examples), testing the robustness of prefix to noise, and exploring the tradeoff between prefix length and the number of in-context examples under a fixed token budget. Moreover, comparisons with more baselines are needed, such as recent methods that implicitly inject task information via symbolic prefixes ([1][2]), as well as natural language instructions as prefix alternatives, to better situate the proposed approach within ICL and instruction learning.
>
> A: We appreciate the feedback from the reviewer. The "requested changes" are composed of weaknesses; please see our answers to weaknesses.

---

> > ### Comment · Reviewer_5Gsi · 2025-09-04
> >
> > Regarding the "Need more ablation" section, I acknowledge your statement that you will consider adding these experiments. I strongly encourage you to conduct these ablations and incorporate the results into the final version, as they are crucial for validating the robustness and core mechanics of your proposed framework.
> > Regarding the "Need more baseline" section, your rebuttal is not convincing.
> > 1. For the comparison with methods [1, 2], I am confused by the claim that a direct comparison is "not feasible" due to different frameworks. It should be possible to conduct a numerical comparison on at least one shared task. In fact, if your method demonstrates superior performance in such a setting, it would significantly strengthen your paper's claims about its relative advantages.
> > 2. Furthermore, your response completely failed to address my second suggested baseline: using "natural language instruction as prefix". It makes me question whether my review was carefully read.
> > Therefore, my initial concerns about the paper's empirical validation remain unaddressed.

---

> ### Author Response · Authors · 2025-09-08
> **About "natural language instruction as prefix"**
>
> Dear Reviewer,
>
> For your followup question 1: our framework works on the task "identifying the hypothesis class". [1] works on identifying the the task (a,b,op) from a task pool specified by instruction. [2] works on linear regression. These three have no overlapping task, therefore I want to claim a direct comparison is not feasible. (We are fully aware of these two papers and they are discussed in our section related work of the submitted version)
>
> For your followup question 2: Our framework is on synthetic dataset and the model is trained from scratch. I'm not able to apply "Natural language instruction as prefix" as baseline. Could the reviewer provide more instruction on how to apply "natural language instruction as prefix" under our synthetic setting? Note that the trained model under synthetic setup has no tokenizer of natural language. Or do I misunderstand what the review mean?

---

### Review · Reviewer_sfsQ · 2025-07-21

**Summary Of Contributions:**

This paper proposes in-context learning with hypothesis-class guidance (ICL-HCG) to study the effect of instructions in ICL. The authors implement many experiments to study several aspects of this setting, including the in-domain and out-of-domain generalization performance, sample complexity, and how model structure, data imbalance, instructions, and the diversity of pretraining hypotheses affect the performance.

**Audience:**

Yes

**Claims And Evidence:**

Yes

**Requested Changes:**

1. It is better to at least add some discussion or simple experiments to further explain the findings in Section 4. For example, why do Transformers and Mamba behave differently?

2. In Figures 11 and 12, what does "position in in-context sequence" mean? Why not have a discussion about how this factor affects the performance?

**Strengths And Weaknesses:**

Strengths:

1. The motivation of this paper is clear and significant.

2. The experiments are extensive, which leads to many conclusions.

Weaknesses:

1. There is no theoretical analysis or understanding in the paper. For example, it is not surprising that Transformers can solve these tasks. But by what constructions, like how many layers are needed, a Transformer is expressive enough to handle these tasks is more interesting.

2. It is better to introduce how the studied synthetic data model is related to real-world applications.

---

> ### Author Response · Authors · 2025-08-25
>
> > There is no theoretical analysis or understanding in the paper. For example, it is not surprising that Transformers can solve these tasks. But by what constructions, like how many layers are needed, a Transformer is expressive enough to handle these tasks is more interesting.
>
> A: The authors ackowledge this limitation. For the scope of this paper, the authors bring the instruction to the ICL community, and provide diverse experiments to motivate many interesting research questions such as the one point out by the reviewer.
>
> > It is better to introduce how the studied synthetic data model is related to real-world applications.
>
> A: We appreciate the feedback and will add more text for this connection. We bring a connection here: An example is that an instruction of "classifying the dairy into the following five categories: very happy, happy, neutral, sad, and very sad". With this instruction, we still need some ICL examples to detail the boundaries of those categories.
>
> Requested Changes:
> > It is better to at least add some discussion or simple experiments to further explain the findings in Section 4. For example, why do Transformers and Mamba behave differently?
>
> A: The authors appreciate the feedback and acknowledge to make revisions accordingly.
>
> > In Figures 11 and 12, what does "position in in-context sequence" mean? Why not have a discussion about how this factor affects the performance?
>
> A: It means the number of ICL samples before the current query. The authors appreciate the feedback and acknowledge to make revisions accordingly.

---

### Review · Reviewer_vP2E · 2025-07-29

**Summary Of Contributions:**

This paper considers studying in-context learning (ICL) with additional information about the hypothesis class. Compared to previous works studying simple function classes that input only $(x, y)$ pairs and the final $x$, the authors concatenate the information of potential hypotheses previously to those $(x, y)$ pairs. Specifically, a hypothesis class $\mathcal{H}$ that contains the true $h$ such that $y = h(x)$ is attached to the input. Under this framework, the authors conduct different numerical experiments to explore the behaviors of in-context learning with hypothesis-class guidance (ICL-HCG).

**Audience:**

Yes

**Claims And Evidence:**

Yes

**Requested Changes:**

1. I recommend that the authors spend more effort in motivating ICL-HCG. Why do you study this framework? It's still not sufficiently addressed. Besides, the analogy to user instructions can be improved or replaced.

2. I suggest that the authors should spend more effort in comparing their experimental findings with existing literature in the main text. I think maybe the authors can add a paragraph after each finding. Why is ICL-HCG special compared to previous settings? What's the difference of LLM behaviors in ICL-HCG?

**Strengths And Weaknesses:**

Strengths:

1. The authors propose a novel type of ICL task input structure.

2. The authors conduct extensive experiments to explore the ICL-HCG task.

Weaknesses:

1. The analogy to "users will provide instructions along with the example" is suspicious: When users clearly instruct the model to perform some task, the performance of the model may rely heavily on the supervised fine-tuning (SFT) result of training the model to do as instructed rather than infer the task from the examples; but the latter one is what ICL cares about.

2. The ICL-HCG framework is neither practical nor insightful. In practice, it's hard to believe that users will provide multiple task instructions and let the model guess which one is correct by constructing some further examples. From the aspect of understanding the ICL phenomena of LLMs, the ICL-HCG is too easy and not insightful. It's restricting the task in a finite (and moderately small) hypothesis class, which is even easier than the original setting. I don't quite see the point of verifying that the LLMs *can* perform ICL under this setting or perform ICL *better* than those without this additional information.

3. The experimental findings are mostly already verified by previous works focusing on simple function classes (e.g., linear regression) without hypothesis class information. It's not surprising to see those findings again in ICL-HCG. Furthermore, the authors do not provide an in-depth and convincing analysis of those phenomena.

---

> ### Author Response · Authors · 2025-08-25
>
> >The analogy to "users will provide instructions along with the example" is suspicious: When users clearly instruct the model to perform some task, the performance of the model may rely heavily on the supervised fine-tuning (SFT) result of training the model to do as instructed rather than infer the task from the examples; but the latter one is what ICL cares about.
>
> A: The authors agree that the latter part is what ICL cares about. The author would like to argue that the analogy is not suspicious. In industry, ICL is usually used with instructions, where the instruction provides a vague constraint on that task and ICL examples provide more nuanced constraints. An example is that an instruction of "classifying the dairy into the following five categories: very happy, happy, neutral, sad, and very sad". With this instruction, we still need some ICL examples to detail the boundaries of those categories. The above case can be based on a pretrained LLM before any finetuning.
>
> >The ICL-HCG framework is neither practical nor insightful. In practice, it's hard to believe that users will provide multiple task instructions and let the model guess which one is correct by constructing some further examples. From the aspect of understanding the ICL phenomena of LLMs, the ICL-HCG is too easy and not insightful. It's restricting the task in a finite (and moderately small) hypothesis class, which is even easier than the original setting. I don't quite see the point of verifying that the LLMs can perform ICL under this setting or perform ICL better than those without this additional information.
>
> A: [Practical] Consider the example we discussed in the last paragraph, the instruction does not write the functions out but implicitly expresses those functions. The authors acknowledge the gap between our framework and a real use case. In other words, the user will provide an explicit multi-task instruction rather than an implicit multi-task instruction. [Insightful] The easiness is where we start to provide insight. The authors believe the core collaboration between instruction and ICL happens in the function space, where the instruction offers a function space, and ICL examples further choose the right function. We hope our simple framework makes this clear to readers rather than hiding the insight under a complex framework or math. Based on the simple setting, we believe our experiments are insightful. For example, in the experiments of OOD hypothesis class generalizations, there is no overlapping between testing and training distributions. But the model can still perform well on testing samples.
>
> >The experimental findings are mostly already verified by previous works focusing on simple function classes (e.g., linear regression) without hypothesis class information. It's not surprising to see those findings again in ICL-HCG. Furthermore, the authors do not provide an in-depth and convincing analysis of those phenomena.
>
> A: In the framework of ICL-HCG, the most basic experiment is the generalization of identifying a function in the instruction. This is already fundamentally different from all the (ICL) works focusing on simple classes, except for the two works listed in Related Work in this paper. Therefore, those findings are not verified by previous works. The authors believe verifying these settings is informative. The reason is that before verifying, one cannot predict whether such generalization will happen or not.
>
>
> Requested Changes:
> >I recommend that the authors spend more effort in motivating ICL-HCG. Why do you study this framework? It's still not sufficiently addressed. Besides, the analogy to user instructions can be improved or replaced.
>
> A: [Motivation] As discussed in our example from the industry, the real use case of ICL is usually collaborated with instruction. Existing literature on ICL usually ignores the instruction; therefore, we hope to bring this to he community. This is a strong motivation. [Analogy] The authors acknowledge the gap between this paper's explicit finite multi-task instruction and real-world implicit potentially infinite multi-task instruction.
>
> >I suggest that the authors should spend more effort in comparing their experimental findings with existing literature in the main text. I think maybe the authors can add a paragraph after each finding. Why is ICL-HCG special compared to previous settings? What's the difference of LLM behaviors in ICL-HCG?
>
> A: We appreciate the feedback from the authors. We will add a paragraph to distinguish our work from previous works to improve the clarity of our contributions to readers.

---

> > ### Comment · Reviewer_vP2E · 2025-09-04
> >
> > Many thanks for your detailed explanations. Looking forward to your corresponding updates in the future version.

---

> > > ### Author Response · Authors · 2025-09-08
> > > **More clarification on Motivation**
> > >
> > > Dear Reviewer,
> > >
> > > We want to provide more content of our **motivation** on instruction:
> > >
> > > (1) The role of instruction. An instruction is used either [to define the task such as "Do the following addition task on two intergers, reply only with the answer" which is super precise and defining the exact task] or [to give the range of the task such as "Act as a mathematician and solve the following math task" which constraints the task]. Our framework considers the second scenario.
> > >
> > > (2) The collaboration of instruction and ICL. As we empirically know instruction+ICL performs better than either instruction or ICL. We believe this is because instrucion provides a hypothesis class of the task, and ICL samples further filter out wrong hypotheses. As ICL is heavily studied by the literature, we use our framework examine such scenario (compare instruction+ICL with ICL).
> > >
> > > We hope the motivation could be super clear to the reviewer!

---

### Decision · Action_Editor_zgBd · 2025-09-27

**Recommendation:** Reject

**Audience:**

Yes

**Audience Explanation:**

As noted above, I am not concerned with audience interests, since indeed a similar setting of instructions + examples in ICL has been studied previously, including in the most related works Huang & Ge (2024) and Xuanyuan et al. (2024).

**Claims And Evidence:**

No

**Claims Explanation:**

This submission studies in-context learning with side information, where a model is given a natural language description of the task alongside in-context examples. While reviewers had no concerns about the correctness of the paper, they maintained primary concerns about (i) the practical relevance of the setting and (ii) the non-existence of baselines.

The discussion was relatively limited on both the authors' and reviewers' sides, so I took a detailed look into assessing both (i) and (ii). I am not concerned with (i), since indeed a similar setting of task description + examples in ICL has been studied, including in the most related works Huang & Ge (2024) and Xuanyuan et al. (2024).

However, I found that the present version of the submission does not sufficiently address its differences from the prior works of Huang & Ge (2024) and Xuanyuan et al. (2024), which also study synthetic ICL tasks with instructions as side information. In particular, the side information in this work is a subset of the full hypothesis space of possible binary classifications on the data, termed a "hypothesis prefix." While this instantiation is different than the prior work (who provide the ground truth mean and sounds bounds on the ground truth, respectively, in simple mathematical tasks) in terms of the motivation of this work as studying "instructed ICL," it is not sufficiently different.

Indeed, after briefly discussing these two works, the submission goes on to state:

> "While prior studies ... train Transformers for ICL on sequences ... without explicit instructions, our work investigates whether a Transformer trained for ICL with instructions..."

thereby ignoring the relationship to Huang & Ge (2024) and Xuanyuan et al. (2024).

The submission wrote about these two prior works in the introduction:

> "... two recent works (Xuanyuan et al., 2024; Huang & Ge, 2024) adopt prefix as instruction to implicitly provide information on the task. In contrast, our approach explicitly provides a hypothesis class as a prefix to the Transformer, guiding the model's understanding of the intended task."

However, I find this distinction insufficient. All these settings guide the model's understanding of the intended task and are meant to simulate the natural setting targeted in the submission: "users often provide LLMs with an instruction in addition to labeled demonstrations, containing the descriptions of the task in mind."

To address this mismatch, the authors could submit a revision that either:

1. **Restricts the claims** to be about hypothesis prefixes only. However, the submission would lose its high-level motivation as targeting the ICL with instructions setting, and thus lose its broad appeal.

or

2. **Expands the class of instructions** considered as side information to include prior work in addition to the types of instructions considered here (hypothesis prefixes), thereby meeting a the bar for follow-up work on the setting of ICL with instructinos.

##### Minor

Given that there is extensive work before the in-context learning era on side/privileged information and task descriptions, I am surprised the submission does not cite these works, and in fact does not cite much work before 2020 (besides the LSTM and GRU papers, and a paper on which the task setup is strongly influenced, Zhu (2015)). Given that similar ideas have already been explored in other contexts, I encourage the authors to conduct a more comprehensive literature review; *e.g.*, this work and works citing it: [Vapnik & Vashist (2009)](https://doi.org/10.1016/j.neunet.2009.06.042).

**Resubmission Of Major Revision:**

The authors may consider submitting a major revision at a later time.